# On demand laser-induced frequency tuning of coherent magnons in a nanometer-thick magnet at room temperature

Volker Wiechert [1], Hanchen Wang [2], William Legrand [2], Pietro Gambardella [2], David Breitbach [3], Philipp Pirro [3], Michaela Lammel [1], Andrea Meo [4], Giovanni Finocchio [5] & Davide Bossini [1] ✉

The collective vibrational and magnetic response of a solid to external stimuli is encoded in the dispersion relations of phonons and magnons, respectively. Recently, the coherent drive and nonlinear manipulation of collective lattice and magnetic excitations via laser pulses has been explored, as a route to control the non-equilibrium properties of quantum materials. Device concepts that leverage coupled multiphysical dynamics must exhibit laser-induced frequency tunability controllable through external parameters. Although previous works have shown that optically driven excitations in the midinfrared can manipulate magnon frequencies, inducing deterministic red- or blue-shifts of the magnon frequency in the same material is still elusive. Here we demonstrate this concept in a nanometer-thick magnet at room temperature. Visible light pulses in combination with an external magnetic field (<200 mT) can either raise or lower the magnon frequency by up to 40% of its original value. This effect results from the interplay of the optical excitation, magnetic anisotropy and external magnetic field. Our results show how an efficient manipulation of magnons can be achieved by light and provide perspectives for the realization of logic devices optically reconfigurable on the nanosecond timescale.

Experiments have recently demonstrated that driving coherent collective excitations in solids leads to macroscopic coherent phenomena[1–6], such as magnetoelectricity[6,7], ferroelectricity[5,8], and magnetism[2,9]. In addition, intense resonant excitation of phonons and magnons activate nonlinear couplings between several modes in the dispersion relations of a given material[10–14]. Developing a method to arbitrarily modify the dispersion relations with light would unlock an even higher level of control. In this scenario the frequency, amplitude, lifetime and even propagation properties (i.e., curvature of the dispersion relation) could be tailored at will by means of laser pulses. In the case of a magnetic material, realizing this goal would have also technological impact in spin-based technologies such as magnonics and spintronics[15,16].

Several experiments have so far reported laser-induced modifications of zone center magnon frequencies at the Γ point. The typical approaches are either photoinducing a magnetic phase transition[6,17] or resonantly driving phonons[18] or magnons[19]. These impressive results were obtained by exciting phonons and magnons in the mid-infrared

[1]Department of Physics and Center for Applied Photonics, University of Konstanz, Konstanz, Germany. [2]Department of Materials, ETH Zurich, Zurich, Switzerland. [3]Fachbereich Physik and Landesforschungszentrum OPTIMAS, Rheinland-Pfälzische Technische Universität Kaiserslautern-Landau, Kaiserslautern, Germany. [4]Department of Electric and Information engineering, Politecnico di Bari, Bari, Italy. [5]Department of Mathematical and Computer Sciences, Physical Sciences and Earth Sciences, University of Messina, Messina, Italy. ✉e-mail: davide.bossini@uni-konstanz.de

spectral range, which is technically demanding[20]. Alternatively the resonant pumping of specific electronic transitions from dopant ions[21] and opto-magnetic effects[22] have been explored. Moreover in several cases bulk materials and cryogenic conditions[6,17,18] (to explore the phase diagram), were necessary. Whether these concepts can be realized at room temperature and scaled to nanometer-thick magnets, both necessary conditions for a technological implementation, is an open question. In particular, operating at room temperature rules out photoinduced phase transitions as a means to modify the magnon spectrum. An essentially different and general concept is therefore required.

Here, we show that visible laser pulses enable a quasi-instantaneous (on magnon timescale) deterministic tuning of the eigenfrequency of coherent magnons in a nanometer-thick magnetic material at room temperature. The frequency tuning can be continuously controlled by up to 40% of the original value. The observed behavior arises from bismuth-substituted yttrium iron garnet (BiYIG)'s low damping, strong magneto-optical response, and the low thermal conductivity of the GSGG substrate, distinguishing it from conventional YIG/GGG systems[23,24]. The interplay among the optical excitation properties, the magnetic anisotropy of the material and the externally applied magnetic field determines the frequency tunability. More

specifically, the magnetic field determines the magnetic state prior to the photoexcitation. Depending on the intensity of the laser excitation, two different regimes of spin dynamics, either anisotropy-dominated or field-dominated, can be photoinduced giving rise to a deterministic magnon red- or blue-frequency shift, respectively. These observations are properly reproduced by means of atomistic simulations of optically driven spin dynamics, in which laser-induced heating is encoded as source term of the magnetic equation of motion. Our results pave the way for the design of the next generation of spintronic and magnonic devices with a dynamical response characterized by multi-physics phenomena.

## Results

Our material of choice is BiYIG[25]. We investigate an ultra-thin (thickness: 20 nm) single-crystal specimen grown (see Methods) on a paramagnetic GSGG substrate. Below the ordering temperature ($T_C \approx$ 580 K) two non-equivalent iron sublattices couple anti-ferromagnetically, giving rise to a ferrimagnetic ground-state (see Fig. 1a). Focusing on the lowest energy magnon band, we can treat BiYIG as a ferromagnet in an effective description. Although lattice-matched BiYIG crystals generally display in-plane anisotropy, the strain induced by the lattice mismatch with the substrate forces the

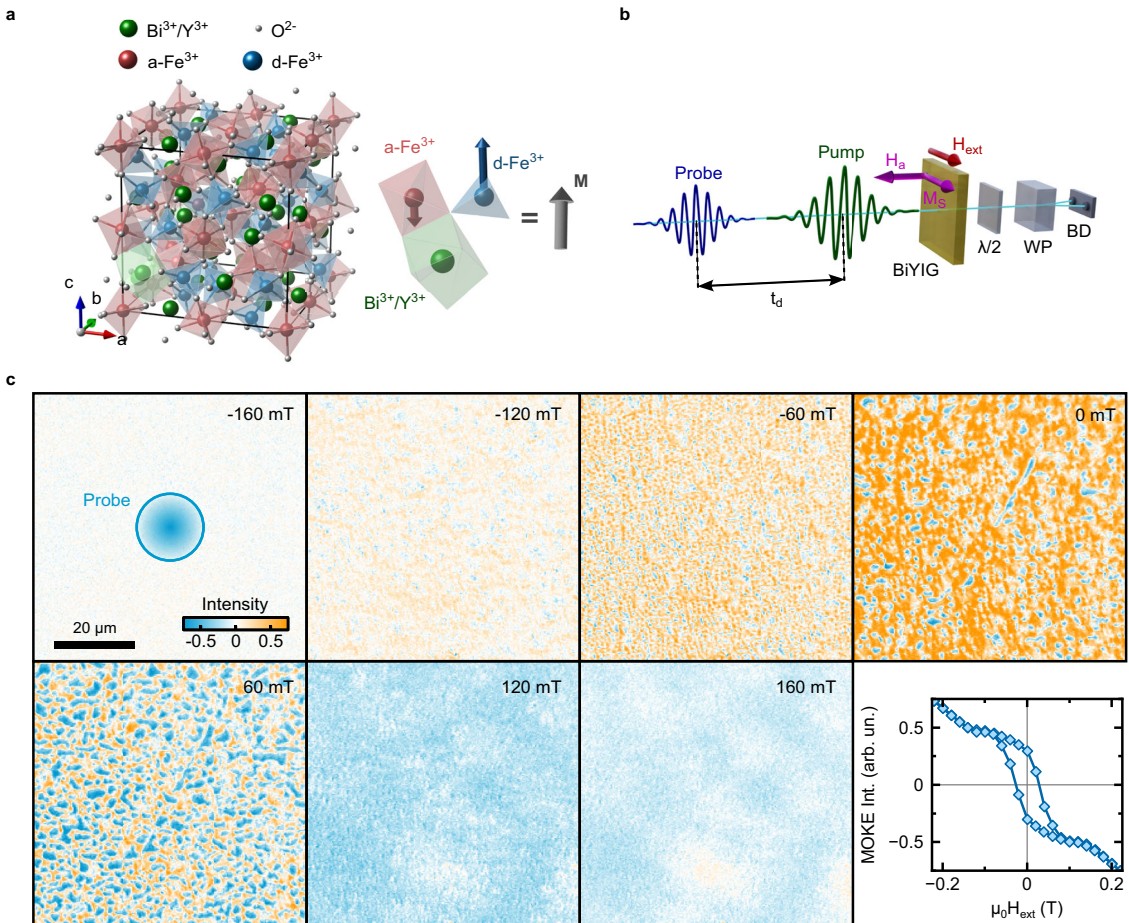

**Fig. 1 | Sample characterization and experimental scheme. a** The crystal structure of rare-earth iron garnets consists of three distinct coordination environments for the cations. Each formula unit contains three Bismuth/Yttrium ions $Bi^{3+}/Y^{3+}$ that are dodecahedrally coordinated with oxygen, three a-$Fe^{3+}$-ions in octahedral coordination, and two d-$Fe^{3+}$ ions in tetrahedral coordination. The finite magnetization arises from the antiparallel alignment of the sublattice magnetizations of the octahedrally and tetrahedrally coordinated iron atoms, which are not equal in magnitude. The minimal magnetization of the $Bi^{3+}$-ions is negligible[45]. **b** Schematic

representation of the two-color pump-probe setup. An external magnetic field is applied to the sample. The probe pulses interact with the sample at a continuously variable delay $t_d$, and the induced polarization rotation is measured using the balanced detection scheme. **c** Magneto-optical Kerr effect (MOKE) imaging measurements as a function of the magnetic field. The hysteresis loop is extracted from the imaging data for external in-plane magnetic fields. The scale bar corresponds to 20 μm. As a reference, the first in (**c**) shows the focal spot size of the probing beam from the pump-probe experiments.

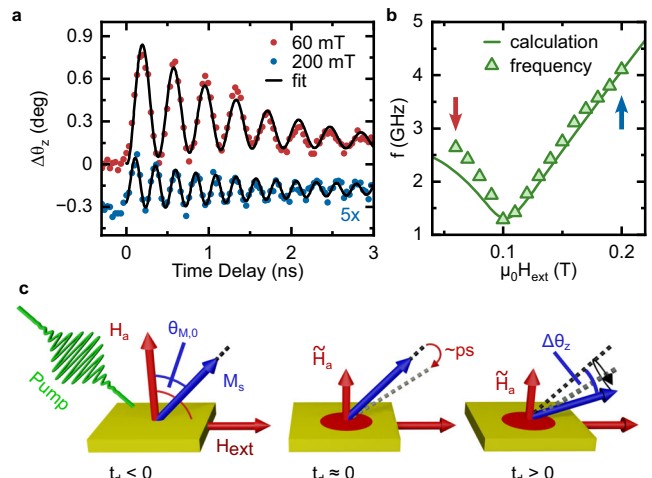

**Fig. 2 | Spin dynamics and excitation mechanism. a** Time-resolved rotation of the probe polarization in the presence of an in-plane external field amounting to 60 mT (red) and 200 mT (blue). The excitation fluence is set to 0.5 mJ/cm². The blue curve is multiplied by a factor 5 for the sake of presentation. **b** Frequencies of the coherent oscillations as a function of the external magnetic field. The values corresponding to the time traces in (**a**) are shown by arrows. The solid line is a calculation from the Kittel model for the ferromagnetic resonance in a thin film (see Methods), revealing the magnetic origin of the oscillations[37]. **c** Excitation mechanism of the coherent magnons by ultrafast reduction of uniaxial anisotropy, as explained in detail in the main text. The precession angle $\Delta\theta_z$ describes the amplitude of the precession. All data set in the time-domain are expressed in terms of $\Delta\theta_z$.

magnetization of our samples to be out-of-plane in the ground state, due to magnetoelasticity[26,27]. As a result, our specimen shows perpendicular magnetic anisotropy (PMA), as demonstrated by the SQUID measurements (Supplementary Fig. 1, see Methods). The literature reports that PMA garnets can be grown for thickness below 200 nm[25,26]. First, we perform magneto-optical imaging of the ground state of our sample (Fig. 1c). These measurements lead to the conclusion that a magnetic field on the order of 100 mT saturates the magnetization in the plane of the sample. Considering the direction and intensity of the external field, the magnetic configuration of the sample can be determined by means of the self-consistency condition in Eq. M1 (Methods section).

Aiming at manipulating coherent collective magnetic excitations, we choose to perform pump-probe measurements with femtosecond temporal resolution. The photon energy (2.4 eV) of the excitation beam enables us to drive electrons above the bandgap of BiYIG, while the probe photon energy (2.9 eV) was tuned to a resonance of the magneto-optical spectrum[28–30]. Our experimental apparatus (Fig. 1b) is sensitive to the transient rotation of the probe polarization and, thus, to the transient Faraday effect (further details of the set-up are in the Methods section). All the experiments discussed in our work are performed at room temperature.

The photoinduced coherent spin dynamics, under the application of a magnetic field with different values, is shown in Fig. 2a. The data are expressed in terms of the precession angle of the magnetization $\Delta\theta_z$, defined in Fig. 2c. The datasets shown in Fig. 2a were obtained for the same excitation fluence. We observe that the amplitude of the coherent magnetic oscillation is remarkably higher in the case of the 60 mT measurement, in comparison with the data acquired under the application of a 200 mT field. The excitation mechanism underlying our observations can be identified as displacive excitation of coherent magnons (DECM)[31–35], since the excitation photon energy exceeds the band gap of the sample. The DECM process, schematically illustrated in Fig. 2c, is triggered by laser-induced heating, which transiently

reduces the uniaxial anisotropy of the material. As a result, the equilibrium orientation of the magnetization is displaced, initiating a precessional motion at the ferromagnetic resonance frequency (FMR) around this new, quasi-equilibrium position. Once heat has dissipated, the system gradually returns to its initial equilibrium state, restoring the uniaxial anisotropy. Other authors have reported a light-induced modification of both the magnitude and direction of the magnetic anisotropy[21]. This effect induces a dependence of the phase of the coherent oscillations on the polarization of the pump beam. We rule out this scenario in view of Supplementary Fig. 2, which proves our mechanism to be pump polarization-independent consistently with the DECM picture. The DECM mechanism also provides an explanation for the observed variation in the oscillation amplitude. In the case of an external magnetic field of 200 mT, the magnetization is saturated within the film plane. Consequently, a reduction in anisotropy induces only a minimal shift in the equilibrium position, resulting in oscillations with a low amplitude (Fig. 2c). In contrast, for an external field of 60 mT, the equilibrium magnetization is pulled out of the film plane. In this configuration a reduction in anisotropy leads to a more pronounced change of the equilibrium orientation, giving rise to a larger oscillation amplitude (Fig. 2c). A systematic variation of the external field enables a transition from the out-of-plane to the in-plane orientation of the magnetization. Correspondingly, a precise tuning of the magnon eigenfrequencies, which are extracted via time-domain fitting, is observed (Fig. 2b). This data set was taken by setting the fluence (F) to a rather limited value (0.5 mJ/cm²) to minimize the perturbation of the sample. We calculate the FMR frequency for different external fields, based on a standard model for magnetic thin film with PMA[36–38]

$$f = \frac{1}{2\pi}\sqrt{\omega_H\left(\omega_H + \omega_M \sin^2(\theta_{M,0})\right)}, \tag{1}$$

where

$$\omega_H := \gamma H_{ext}\cos(\theta_{M,0} - \theta_H) - \omega_M \cos^2(\theta_{M,0}) \text{ and } \omega_M := \gamma\mu_0 M_{eff}. \tag{2}$$

The symbol $H_{ext}$ describes the externally applied field, the angles $\theta_{M,0}$ and $\theta_H$ are defined in Fig. 2c, $M_{eff} := M_S - H_a$ ($M_S$ is the saturation magnetization and $H_a$ the effective anisotropy field), $\gamma$ is the gyromagnetic ration and $\mu_0$ the vacuum permeability. The calculations (details in Methods) exhibit excellent agreement with the experimental data if the magnetic field exceeds 100 mT (Fig. 2b). This field regime corresponds to a saturated magnetization in the sample plane (Fig. 1c). Differently, calculations reproduce the data less closely if the magnetic field is weaker than 100 mT (Fig. 2b). In this physical regime the sample is in a multi-domain state (see Fig. 1c), which is not taken into account in the employed model. The conventional approach to all-optical experiments avoids this physical regime. According to the commonly accepted wisdom, focusing laser pulses on an area containing several domains does not result in observing coherent spin dynamics. Since the magnonic oscillations possess different phases in the different domains, overall oscillations with a well-defined phase are expected to be averaged out. Moreover, other phenomena related to the presence of domains, such as splitting of the FMR frequency are typically expected[39]. None of these predictions find confirmation in Fig. 2a. We understand the data in multi-domain state considering that the in-plane field biases the local oscillation axis of the magnetization to a specific direction. Consequently, coherent signal from different domains with the same magnetization direction dominates on the signal generated by domains, in which the magnetization is differently oriented. The magnetic field-dependence of the magnon amplitude (Supplementary Fig. 3b) confirms this statement, showing that as the in-plane field is reduced, the signal amplitude decreases and approaches zero for a vanishing field. Additional measurements performed

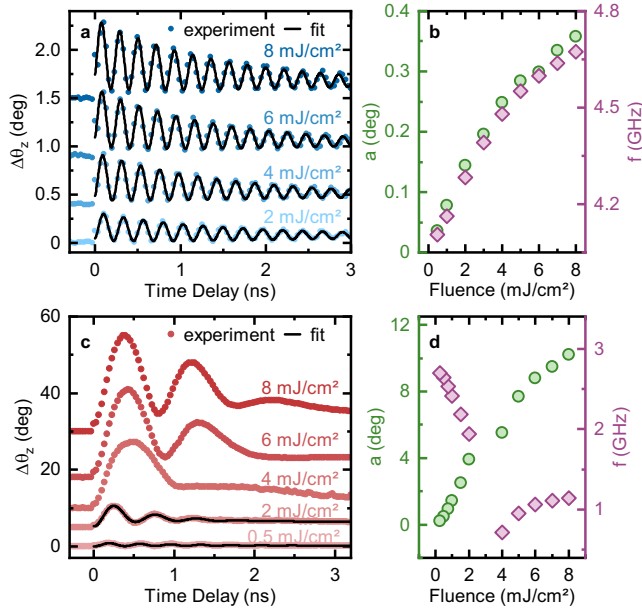

**Fig. 3 | Deterministic frequency up- and down-conversion. a** Fluence dependence of the time-resolved rotation of the polarization in the presence of an external field amounting to 200 mT. The solid lines represent the corresponding fits by using Eq. S1. **b** Precession angle and oscillation frequency extracted from a time-domain fit of the data in (**a**). **c** Fluence dependence of the time-resolved rotation of the polarization in the presence of an external field amounting to 60 mT. The solid lines represent the corresponding fits by using Eq. S1. As noted in the text, for the high-fluence regime (i.e., above 2 mJ/cm²) the lifetime of the oscillations is strongly suppressed. This fact forbids a successful fit to the data with a harmonic oscillating function (see Eq. S1). We thus rely on the Fourier transform to extract the amplitude and frequency of the oscillating component of the signal. **d** Precession angle and oscillation frequency extracted from the data in (**c**).

sweeping the field from − 200 to 200 mT and back again to −200 mT (Supplementary Fig. 4) reveals that the sign of the field and the direction of the field sweep have no systematic detectable effect on the frequency and amplitude of coherent magnons. Measurements of the spin dynamics as a function of the excitation fluence are shown in Fig. 3a ($\mu_0 H_{ext} = 200$ mT). The extracted oscillation amplitudes and frequencies are presented in Fig. 3b. These curves reveal that both the amplitude and frequency scale with the intensity of the optical drive. This effect has been previously attributed to a magnetoelastic modification of the uniaxial anisotropy, induced by the optical generation of long-lived (nanosecond timescale) strain waves[40,41]. The effective value of the uniaxial anisotropy in the photoexcited state can be retrieved, by fitting the data obtained as a function of the magnetic field for each value of the pump fluence (see Methods and Supplementary Fig. 5). The laser-induced demagnetization (see Methods) was taken into account in the fits. The excellent quality of the data fitting reported in Supplementary Fig. 5 shows that a modification of the uniaxial anisotropy can indeed explain the observed blueshift of the magnonic frequency. We observe that the magnon damping depends on the laser fluence (see Methods and Supplementary Fig. 6b). It increases linearly up to 5 mJ/cm² before saturating. This behavior is attributed to laser-induced incoherent phonons that scatter with spin waves, reducing their lifetime until a bottleneck occurs near this fluence. A microscopic explanation is beyond the scope of the present manuscript.

Next, we analyze the magnetization oscillations obtained under an external field of 60 mT. As depicted in Fig. 3c, the oscillation amplitude exhibits a pronounced increase even at low excitation intensity, i.e., up to $F = 2$ mJ/cm². If the fluence is set to 2 mJ/cm², the magnon amplitude exceeds by an order of magnitude the oscillations observed for the

same pump intensity with a 200 mT field (Fig. 3a). In this limited fluence regime the data can be satisfactorily fitted by means of Eq. S1 in the time-domain. Further increasing the excitation (i.e., $2 < F \leq 8$ mJ/ cm²) reveals a dramatic change in the dynamics: the amplitude of the oscillations increases strongly, while the lifetime is suppressed, similar to the situation observed for metallic thin films[34]. The transition from one physical regime to a different one is witnessed by the evolution of the amplitude and frequency, observed by ramping up the fluence (Fig. 3d). Importantly, the frequency is redshifted by 40% of its original value as the fluence is enhanced up to 4 mJ/cm², while it increases for even higher excitation intensity. Hence the data demonstrate that, by tuning the magnetic field between 60 and 200 mT, we can select the sign of the laser-induced frequency modification of the magnonic eigenfrequency.

We move now to the discussion and interpretation of this key result of our work. The literature reports nonlinear magnonic phenomena in rare-earth-doped garnets triggered by microwave excitation[42,43]. In particular, the microwave-induced magnon population is so high that the frequency is renormalized, thus becoming a function of the magnon population. It makes therefore sense to wonder whether this nonlinear effect is responsible for the frequency tuning observed in our laser-driven system. We rely on the analytical theory for the nonlinear spin-wave frequency shift[37] and calculate the magnitude of this effect in our experimental configuration (see Methods). We obtain that the predicted frequency self-shift induced by the observed magnon intensities is below $|\Delta\omega_0^{max}|/2\pi = 0.016$ GHz for the precession amplitudes observed in the experiment. Therefore, this effect provides only a minor contribution and cannot be responsible for the detected red- and blue-shifts.

We thus provide the following physical picture for the data in Fig. 3. Applying a field of 60 mT does not saturate the magnetization in the sample plane. Therefore, an out-of-plane component of the magnetization, determined by the PMA, is still present. As already mentioned, the optical excitation weakens the anisotropy. Since the precession frequency depends on the anisotropy (Eqs. M2–M3), the continuous curve in Fig. 2b, and its minimum with it are shifted. A reduction of the anisotropy effectively shifts the curve towards lower magnetic field values (i.e., to the left in Fig. 2b). However, although weakened, the anisotropy is still the most relevant magnetic interaction in the low-fluence regime (anisotropy-dominated). Once the optical pumping is strong enough to overcome the anisotropy, the external magnetic field becomes dominant (field-dominated regime), aligning and saturating the magnetization in the plane of the sample. This phenomenon explains the transition between two different regimes observed in Fig. 3d: Setting $F \geq 4$ mJ/cm² provides a comparable increasing fluence dependence of the frequency, as observed in saturation (Fig. 3b). In both experiments the field dominates on the PMA in the observed spin dynamics. This picture is consistent with the pronounced precession angle observed in the oscillations for $F \geq 4$ mJ/ cm², since in this case a spin-reorientation towards the sample plane is photoinduced.

Seeking a confirmation of our picture, we set out to perform atomistic spin dynamics simulations. The excitation mechanism of the coherent magnons (DECM) is mainly due to the laser-induced heating. Hence, we rely on the two-temperature model to convert the optical excitation into a thermal drive for spins, in the Landau-Lifschitz-Gilbert equation (see Methods). Our experimental results are summarized in Fig. 4a. A two-dimensional plot displays the control on the magnon frequency achieved changing both the magnetic field and the laser fluence. The dependence of the frequency on the fluence for two values of the field (60 mT and 200 mT) is shown in Fig. 4b, while Fig. 4c reports the field dependence of the frequency for the low- (0.5 mJ/cm²) and high-fluence (8 mJ/cm²) regime. The latter plot can be directly compared with the outcome of the simulations (Fig. 4d), which are consistent with the experimental trend, although a higher value of $F$ is

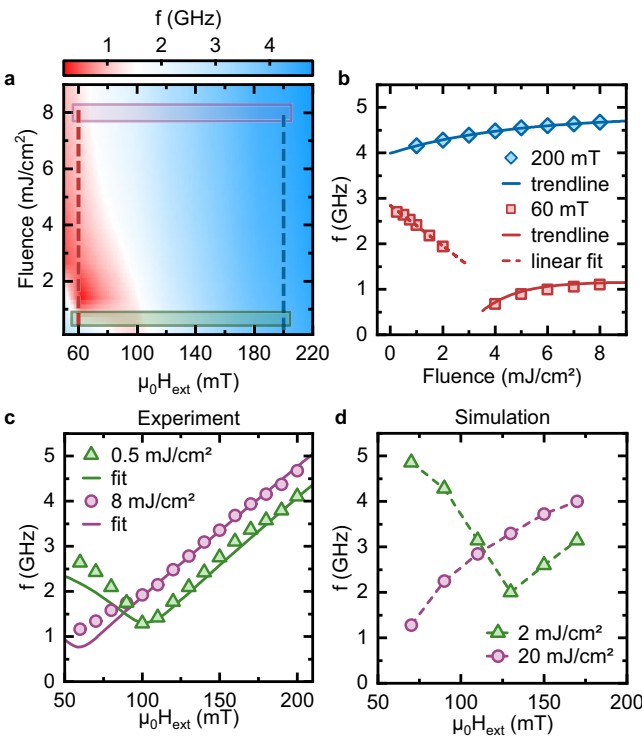

**Fig. 4 | Experimental results and simulations and theoretical prediction.**
**a** Magnon frequency as a function of the magnetic field and pump fluence. The
colored boxes isolate datasets, represented in (**c**). **b** Fluence dependence of the
magnon frequency in correspondence of $\mu_0 H_{ext} = 60$ mT and $\mu_0 H_{ext} = 200$ mT. The
dashed and continuous lines are guides to the eye. **c** Calculated frequency shift
using Eqs. M1–M3 (see Methods) and experimentally observed frequencies as a
function of the in-plane external field corresponding to values of the excitation
fluence equal to 0.5 and 8 mJ/cm². **d** Extracted frequencies as a function of the in-
plane external field from simulations of laser-induced dynamics for fluences of 2
and 20 mJ/cm².

required to reproduce the high-fluence regime ($F = 8$ mJ/cm² in
Fig. 4a–c), dominated by the field. We attribute this discrepancy to two
main factors: I) the two-temperature model adopted here is better
suited to describe metallic systems than insulators, where there is a
further channel to dissipate/excite (i.e., phonons), which may result in
the necessity to provide further energy to the system and, hence,
higher fluence, to reproduce the experimental results. II) We simulate a
system size smaller than the domain size, therefore the obtained
dynamics is related to the quasi-coherent collective response of all the
spins. Nonetheless, the semiquantitative agreement between theory
and experiment further corroborates our previous statement. We
observe that a similar phenomenology of the magnonic frequency
conversion has been recently discussed, concerning DC voltage-driven
magnetic transport in tunnel junctions[44]. It is important to highlight
that the magnetic-field-assisted control of the laser-driven magnonic
frequency shift necessitates a PMA system, so that the external field
can be applied perpendicularly to the magnetic easy axis. The
demonstrated on-demand control of the laser-induced magnon fre-
quency tuning is a landmark for the research field aiming at the optical
coherent control of solids. It hints towards the possibility of coherently
driving instabilities and phase transitions in a magnetic material, by
softening the resonance frequency to zero. Our results pave the way
for the development of optically tunable magnonic devices capable of
operating at room temperature, under the application of moderate
magnetic fields, and using commercially available light sources. This
represents a significant step towards practical implementations of
magnonic technologies. In conclusion, the experimental results and

theoretical modeling of multiphysics control of spin-wave dynamics
provides a robust framework and offers a promising strategy for future
applications in magnonic logic, signal processing, and next-generation
spintronic information technologies.

## Methods

### Sample growth and characterization

Bi-doped YIG epitaxial films are prepared via high-temperature RF
magnetron sputtering, using a stoichiometric target composition of
$Bi_{0.8}Y_{2.2}Fe_5O_{12}$. The substrate is mounted on a planetary holder, which
rotates in parallel alignment to the target and is set 80 mm above it in
an off-axis arrangement. Before deposition, the (111)-oriented
$Gd_3Sc_2Ga_3O_{12}$ (GSGG) substrate is pre-annealed at 750 °C in an Ar-O₂
mixed atmosphere for 1 h. The BiYIG film is then deposited under RF
power of 55 W, with a controlled Ar-O₂ gas mixture flow rate of 100
sccm and 4 sccm, respectively, at a total chamber pressure of
3.8 mTorr. After deposition, an additional annealing step is applied for
1 hour before the sample is allowed to cool to room temperature. The
magnetic parameters of our samples are: $\mu_0 M_s = 125$ mT,
$\mu_0 M_{eff} = -95.6$ mT, $\mu_0 H_a = 220.6$ mT (introduced in Eqs. M1 und M4), as
determined by superconducting quantum interference device (SQUID)
magnetometry and broadband ferromagnetic resonance[23].

### Calculation of the FMR spectrum

According to the well-established theoretical description of magnetic
films[37], the static magnetization angle $\theta_{M,0}$ (Fig. 2(c)) in equilibrium is
an implicit function of the external field $H_{ext}$, its angle relative to the
film normal $\theta\_H$ and the effective magnetization $M_{eff}$:

$$H_{ext}\sin(\theta_{M,0} - \theta_H) = \frac{M_{eff}}{2}\sin(2\theta_{M,0}), \text{ where } M_{eff} := M_S - H_a. \quad \text{(M1)}$$

$M_S$ is the saturation magnetization and $H_a$ the magnetic aniso-
tropy. Having obtained the value of $\theta_{M,0}$ from Eq. (1), the frequency of
the ferromagnetic mode can then be calculated by means of the Kittel-
equation for in-plane magnetized thin films

$$f = \frac{1}{2\pi}\sqrt{\omega_H\left(\omega_H + \omega_M \sin^2(\theta_{M,0})\right)}, \quad \text{(M2)}$$

where

$$\omega_H := \gamma H_{ext}\cos(\theta_{M,0} - \theta_H) - \omega_M \cos^2(\theta_{M,0}) \text{ and } \omega_M := \gamma \mu_0 M_{eff}. \quad \text{(M3)}$$

### Calculation of the nonlinear frequency shift

We calculate the self-nonlinear spin-wave frequency shift following the
Hamiltonian formalism of nonlinear spin-wave dynamics[37]. We con-
sider a magnetic film of thickness $d = 20$ nm, saturation magnetization
$M_S$, exchange constant A (so that the exchange length is $\lambda = \sqrt{2A/(\mu_0 M_S^2)}$,
and an out-of-plane uniaxial anisotropy constant $K_u$, giving an aniso-
tropy field

$$H_a = \frac{2K_u}{\mu_0 M_S}. \quad \text{(M4)}$$

Furthermore, we neglect contributions of cubic anisotropy. Under
the application of an external magnetic field $H_x$ along the in-plane x
axis, we calculate the out-of-plane angle of the static magnetization $\theta_M$
using Eqn. M1. The resulting internal effective field and its magnitude

calculate to

$$\mathbf{H}_{int} = -\frac{1}{\mu_0}\frac{\delta\epsilon}{\delta \mathbf{M}} = H_x\hat{e}_x + (K_u M_z - N_{zz}M_z)\hat{e}_z,$$
$$|H_{int}| = \sqrt{H_x^2 + ((K_u - 1)M_S\cos\theta_M)^2}, \qquad (M5)$$

where we applied the demagnetization tensor element $N_{zz} \approx 1$ for thin films. Using the magnetic thin film functions

$$f(kd) = 1 - \frac{1 - e^{-|kd|}}{|kd|}, \text{ and } F_{zz,k} = 1 - f(kd) - \frac{H_a}{M_S}, \qquad (M6)$$

as well as $\omega_M = \gamma\mu_0 M_S$, $\omega_H = \gamma\mu_0 H_{int}$, we construct

$$Q_k = \frac{\omega_M}{2}\left(2\lambda^2 k^2 + f(kd) + F_{zz,k}\sin^2\theta\right),$$
$$B_k = \frac{\omega_M}{2}\left(f(kd) - F_{zz,k}\sin^2\theta\right), \qquad (M7)$$
$$\Gamma_{zz,k} = \omega_M(\lambda^2 k^2 + F_{zz,k}\cos^2\theta).$$

With $A_k = \omega_H + Q_k$, we can now get the spin-wave frequency $\omega_k = \sqrt{A_k^2 - B_k^2}$. Finally, the spin-wave frequency of mode $k$, shifted by the nonlinear frequency shift at a given spin-wave amplitude $c_k$ is given by

$$\widetilde{\omega}_k = \omega_k + T_k|c_k|^2 = \omega_k + \Delta\omega_k, \qquad (M8)$$

with the self-nonlinear shift coefficient

$$T_k = -Q_k + \frac{B_k^2}{2\omega_k^2}(\omega_H + \Gamma_{zz,2k}) + \left(1 + \frac{B_k^2}{\omega_k^2}\right)\Gamma_{zz,0}. \qquad (M9)$$

This allows us for a direct estimation of the nonlinear spin-wave frequency shift $T_k|c_k|^2$ for a given spin-wave amplitude $c_k$ ($c_k$ is always <1). Using our film parameters and neglecting ellipticity effects on the spin-wave amplitude, we find for the FMR mode ($k = 0$) that the maximum expected nonlinear frequency shift for the spin-wave amplitudes observed in the experiment is on the order of $|\Delta\omega_0^{max}|/2\pi \approx 0.1$ GHz.

| Applied magnetic field $B_x$ (mT) | Max. experimental Precession angle $a$ (deg) | Corresponding spin-wave intensity $|c_0|^2$ | Calculated self-shift coefficient $T_0$ (GHz) | Max. frequency shift $T_0|c_0|^2$ (GHz) |
|---|---|---|---|---|
| 60 | 10.2 | 0.0158 | −1.02 | −0.016 |
| 200 | 0.35 | $1.87 \times 10^{-5}$ | 2.3 | $4.3 \times 10^{-5}$ |

Parameters used for the calculation:

$$\mu_0 M_S = 125 \text{ mT}, d = 20 \text{ nm}, K_u = 10.971\frac{J}{m^3}, \mu_0 B_a = 220.6 \text{ mT},$$
$$A_{ex} = 4.2\frac{pJ}{m}, \lambda = 25.99 \text{ nm}, \gamma = 28.5 \text{ GHz/T}$$

### Laser system and magneto-optical pump-probe experiment

We perform time-resolved magneto-optical experiments with femtosecond temporal resolution using a commercial Yb:KGW laser system at a central wavelength of 1026 nm with a repetition rate of 50 kHz. Frequency doubling the fundamental beam we obtain pump pulses with duration of 230 fs and with photon energy of 2.4 eV (513 nm), nearly resonant with the bandgap of Bi:YIG. To maximize the detection sensitivity, we tune the photon energy of the probe beam to 2.95 eV (420 nm) in correspondence of a maximum of the spectrum of the Faraday rotation[28]. We achieve the tunability of the probe beam by means of a two-stage non-collinear optical parametric amplifier

followed by second harmonic generation. The pulse duration of the probe pulses is 34 fs. The excitation beam is focused to a $1/e^2$-spot size of $23 \times 23\ \mu m^2$, while the probe beam is focused to $15 \times 15\ \mu m^2$, respectively. Both beams used in the experiments of the main text are linearly polarized along the horizontal plane.

## Data availability

Source data are provided with this paper. Further datasets collected for this study are available from the corresponding authors on reasonable request. Source data are provided with this paper.

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

## Acknowledgements

This work was supported by the Deutsche Forschungsgemeinschaft (DFG) through the program BO 5074/1–1 and the grant 425217212 (SFB1432, infrastructure and equipment). The authors thank Stephan Eggert and Christian Beschle for technical support. This work was supported by the collaborative network COST Action "CHIROMAG" (CA 23136). M.L. acknowledges support from the Deutsche Forschungsgemeinschaft (DFG, German Research Foundation) via the grant 25217212 (SFB1432, Project 425217212). P.P. and D.Br. acknowledge funding by the Deutsche Forschungsgemeinschaft (DFG, German Research Foundation) within the Transregional Collaborative Research Center-TRR 173-268565370 "Spin + X" (project B01 and B11). H.W. acknowledges the support of the China Scholarship Council (CSC, Grant No. 202206020091). W.L. acknowledges the support of the ETH Zurich Postdoctoral Fellowship Program (Grant No. 21-1 FEL-48). The work of A.M. and G.F. was supported by Project No. PRIN 2020LWPKH7 funded by the Italian Ministry of University and the Petaspin Association (https://www.petaspin.com).

## Author contributions

V.W. developed the experimental set-up, performed the experiment, analyzed the data and wrote the manuscript under the supervision of D.Bo. H.W., W.L. and P.G. grew and characterized the samples. M.L. performed the characterization SQUID measurements. D.B. and P.P. performed the calculations concerning the nonlinear frequency shift. A.M and G.F. performed atomistic spin dynamics simulations. D.Bo. conceived the project, supervised the experimental activity and the manuscript preparation. All authors took part to regular discussions and contributed to the manuscript preparation. Figs. 1b and 2c were created with *Blender* (Blender Foundation, Amsterdam, Netherlands).

## Funding

## Competing interests

Authors declare no competing interests.
