## [Transparent Peer Review file · Nature Communications]

On demand laser-induced frequency shift of coherent magnons in a nanometer-thick magnet at room temperature

Corresponding Author: Dr Davide Bossini

Version 0:

Reviewer comments:

Reviewer #1

(Remarks to the Author)

The manuscript "On demand laser-induced frequency shift of coherent magnons in a nanometer-thick magnet at room temperature" by Wiechert et al. reports on deterministic tuning of the eigenfrequency of coherent magnons in a nanometer-thick magnetic material at room temperature. The frequency shift can be continuously controlled by up to 40 % of the original value by variation of optical pulse energy. Thereby, depending on the pulse energy two different regimes of spin dynamics, either anisotropy-dominated or external field dominated are established, giving rise to a deterministic magnon red- or blue-frequency shift, respectively. The authors perform large variety of magneto-optical experiments. The data are of high quality and the demonstration of such clean tuning of magnon frequencies in combination with weak damping is impressive. The technical aspects of the manuscript are robust, and the results are convincing. However, the main achievement of the manuscript has to be clarified, as the text lacks the focus on the advancement with respect to previous studies of optically induced ferromagnetic resonance in iron garnet films. This concerns the clarification with respect to the following publications PRB 81, 214440 (2010), PRB B 97, 014422 (2018), New J. Phys. 25, 033016 (2023), and Ref.[38] where heat-induced variation of magnetic anisotropy was shown to trigger ferromagnetic resonance and was employed as a method for tuning the frequency of magnon modes.

Another important aspect concerns the thermal nature of the employed mechanism. Heating can pose a significant obstacle to practical applications, as it limits the operational speed and leads to inefficient energy consumption. This issue is particularly critical in dielectric materials, where cooling is substantially slower than in metals. The observation that the frequency does not change significantly within the first nanosecond indicates that the cooling process is relatively slow. If the tuning of frequency is too slow it can be replaced by other approaches where an external magnetic field is varied to influence the spectrum.

These points should be addressed in the next revision before I can consider recommending the manuscript for publication. Additional questions are listed as well below.

1. The origin of the oscillations is explained as due to a reduction of the anisotropy field. However, in the geometry where the easy axis is strictly out of plane, the external magnetic field is in-plane, and the anisotropy field experiences only reduction (without change of direction), I would expect no signal. The authors mention such a scenario in the multidomain regime but do not resolve the obstacle. Furthermore, even in a monodomain sample, it is not clear which factors define the direction of the anisotropy field if the external field is strictly perpendicular to it.
2. Is the easy axis strictly perpendicular to the plane surface? Do the authors observe hysteresis in magnetization curves for in-plane magnetic field?
3. Is it only reduction of anisotropy or there are changes in its orientation? The authors demonstrate that their signals do not depend on excitation polarization (pump). It is however not clear what happens when the direction of magnetic field changes in-plane with respect to crystallographic axes.
4. What is the role of ultrathin material in this study? What is the size of domains?
5. What is the origin of magnon damping? How strong is the effect of heating on damping?

Reviewer #2

(Remarks to the Author)

Report on manuscript

Authors: V. Wiechert et al.

Title: On demand laser-induced frequency shift of coherent magnons in a nanometer-thick magnet at room temperature

This manuscript reports experimental studies of the photo-excitation of coherent magnons by visible light pulses in films of bismuth-substituted yttrium iron garnet (Bi:YIG) under an external magnetic field applied in the film plane. The magnons are excited by a pump beam with varying fluence of photons with energy 2.4 eV by means of a mechanism known as displacive excitation of coherent magnons (DECM), in the equilibrium orientation of the magnetization is displaced, initiating a precessional motion characterizing a zone-center magnon. The magnon detection is made by a probe beam of photons with energy 2.9 eV tuned to a resonance of the magneto-optical spectrum of Bi:YIG. The detected probe beam exhibits a transient oscillating decaying response from which the magnon frequency is extracted. The authors claim that their results show how efficient manipulation of magnons can be achieved by light and provide perspectives for the realization of tunable spin-wave generators and reconfigurable logic devices. The results are very interesting, the experiments and theoretical interpretation are presented in detail, and the paper is well written. In my opinion the paper will attract attention of the magnetism and spintronics community and deserves to be published in Nature Communications. The only minor suggestion I have is that in the title and text the word tune is better than shift. I understand that shift would apply to a magnon excited with a certain frequency and while it is propagating an external action changes, or shifts, its frequency, as it was shown a long time ago in Frequency Conversion of Spin Waves in Pulsed Magnetic Fields, Applied Physics Letters 10, 184 (1967).

Reviewer #3

(Remarks to the Author)

The authors show the red- and blue-tunability of the magnon frequency in a strained 20 nm BiYIG thin single-crystal. They control the optical excitation fluence and the magnitude of the externally applied magnetic field (to modify the magnetic state before photoexcitation) to transiently alter the uniaxial magnetic anisotropy and magnetization direction, which initiates the precessional motion at FMR. The authors find that in the low-fluence regime, the response is dominated by anisotropy, while in the high-fluence regime, it is dominated by the external field. Atomistic spin dynamics simulations qualitatively capture the experimental findings.

Measurements, data analysis, and simulations follow standard protocols and equations; hence, I find them satisfactory.

The interesting angle the authors use here is the external field regime where magnetization is not saturated and a multi-domain state is present, which leads to red- and blue-shifted responses in the same material, which is not shown before and has the potential for the realization of tunable spin-wave generators and reconfigurable logic devices.

I have a few technical queries:

1. As the authors rightly point out, in a multi-domain state, a coherent signal is not expected due to the random orientation of domains, but a coherent signal is observed here. I am not quite sure why a coherent state is observed. Is the multi-domain state due to an external field that is not quite random, or is the same type of domains dominating the response? The authors must address this point conclusively by mapping the orientation of multi-domain state and also address why it leads to a coherent signal.
2. The authors have also not addressed what happens when we follow the hysteresis curve from -200 mT to 200 mT and then cycle back from 200 mT to -200 mT. Is the response different?
3. The authors must carry out detailed sample characterization, such as magnetization vs T data, strain value, and magnetic moment on both sublattices. Currently, values are used from the literature, but since these values can potentially impact reproducibility, the authors must show these measurements.

Typo: In the methods section below equation M(17), C_p is written as zero, while it should be C_e .

Overall, the results are interesting and presented quite compactly, focusing on just one central point of red-and blue-tunability of magnon frequency. Once the given data's robustness and origin are addressed, the manuscript can be considered for publication in Nature Communications.

Version 1:

Reviewer comments:

Reviewer #1

(Remarks to the Author)

The authors have addressed the main points and revised the manuscript appropriately.

As a minor point, it remains somewhat unclear to me why the authors were able to achieve such low thresholds and outstanding tuning characteristics in their work. Is this mainly related to particular properties of the samples they used? It would be beneficial to include a brief explanation in the text to clarify this point.

Reviewer #2

(Remarks to the Author)

I am happy that the authors accepted my suggestion and replaced the word “shift” by “tuning” in the title of the manuscript and throughout the text. As I stated in my previous report, the results described are very interesting, the experiments and theoretical interpretation are presented in detail, and the paper is well written. Thus, in my opinion the paper will attract attention of the magnetism and spintronics community and deserves to be published in Nature Communications.

Reviewer #3

(Remarks to the Author)

I am satisfied with the author's response to my queries. The manuscript can now be processed for publication.

Reply to Reviewer #1:

The manuscript “On demand laser-induced frequency shift of coherent magnons in a nanometer-thick magnet at room temperature” by Wiechert et al. reports on deterministic tuning of the eigenfrequency of coherent magnons in a nanometer-thick magnetic material at room temperature. The frequency shift can be continuously controlled by up to 40 % of the original value by variation of optical pulse energy. Thereby, depending on the pulse energy two different regimes of spin dynamics, either anisotropy-dominated or external field dominated are established, giving rise to a deterministic magnon red- or blue-frequency shift, respectively. The authors perform large variety of magneto-optical experiments. The data are of high quality and the demonstration of such clean tuning of magnon frequencies in combination with weak damping is impressive. The technical aspects of the manuscript are robust, and the results are convincing.

Comment 1

However, the main achievement of the manuscript has to be clarified, as the text lacks the focus on the advancement with respect to previous studies of optically induced ferromagnetic resonance in iron garnet films. This concerns the clarification with respect to the following publications PRB 81, 214440 (2010), PRB B 97, 014422 (2018), New J. Phys. 25, 033016 (2023), and Ref.[38] where heat-induced variation of magnetic anisotropy was shown to trigger ferromagnetic resonance and was employed as a method for tuning the frequency of magnon modes.

Reply to comment 1

We thank the Reviewer for their careful reading and thoughtful assessment of our manuscript, as well as for highlighting the high quality of the experimental data and the robustness of our results.

The question raised concerning the relation of our results to the literature is highly relevant. As noted by the Reviewer, this aspect was not addressed in enough detail in the original version of the manuscript.

The present study establishes the following advances over prior reports on optically induced magnon frequency control, in terms of frequency tunability, efficiency of the frequency manipulation and nanoscaling of the sample:

1) Earlier works demonstrated magnon frequency tunability relying on polarization-dependent processes, selective resonant pumping of dopant transitions, and measurements restricted to the magnetically saturated regime. Differently our manuscript reports a polarization-independent excitation mechanism that is not tied to dopant-specific transitions. The efficiency of frequency tuning is improved by an order of magnitude, requiring only 4 mJcm^{-2} for a 40 % shift, compared to 80 mJcm^{-2} for 50 % in earlier work (PRB **81**, 214440 (2010)).

2) Our experiments are performed on an ultrathin 20 nm garnet film with a magnetization saturated by a field of 125 mT, one order of magnitude larger than in previous samples (PRB **81**, 214440 (2010), PRB **97**, 014422 (2018)) and grown on common (111)-oriented substrates, in contrast to micrometer thick and compositionally complex films previously required (PRB **81**, 214440 (2010), PRB **97**, 014422 (2018)).

3) The agreement between experiment and theory is also extended. Earlier studies reported only phenomenological descriptions with marginal consistency in limited regimes (PRB **81**, 214440 (2010)), while our work shows excellent correspondence across both saturated and unsaturated conditions, supported by phenomenological and analytical theory as well as numerical simulations.

4) Finally, frequency control is achieved without the need for metallic heterostructures (New J. Phys. **25**, 033016 (2023)), thus avoiding Joule dissipation and establishing a direct route to efficient, on-demand magnon frequency tuning in garnet films.

Below, we summarize the direct comparison with the papers highlighted by the Reviewer. We will then point out the originality and novelty of our results:

Publication	Short summary	Excitation mechanism	Tuning characteristics	Theoretical description
PRB 81 , 214440 (2010)	Photoinduced change of magnon frequency in Co-doped garnet film	 • Polarisation-dependent change of the anisotropy • Effect enabled exclusively by the resonant pumping of electronic transition of a dopant (Co-ions) 	 • 80 mJcm⁻² needed for frequency tuning of 50% • 8 μm-thick sample with almost compensated magnetization, saturated with a field of 9 mT • Separate frequency tuning regimes, no photoinduced transition from “anisotropy-dominated” to “field-dominated” regime, i.e. no on-demand optically induced frequency tuning. 	 • Marginal agreement between theory and experiment in the field regime below saturation • Phenomenological model, no numerical simulations
PRB B 97 , 014422 (2018)	Photoinduced generation of coherent magnons in a 100 μm thick garnet film	Excitation via inverse Faraday-effect (polarization dependent effect) and thermal change of anisotropy	 • No frequency tuning of magnons • Measurements only in saturation • Very specific sample orientation to provide the anisotropy: ferrimagnetic garnet (Y_{0.99}Bi_{1.64}Pr_{0.25}Lu_{0.23})(Fe_{3.75}Ga_{1.16})O₁₂ 10-μm film, grown on the low-symmetry (210)-oriented Gd₃Ga₅O₁₂ 	 • Phenomenological approach based on the Landau-Lifshitz-Gilbert model
New J. Phys. 25 , 033016 (2023)	Optical excitation of coherent magnons in an interface Pt/YIG	Heating of metallic layer inducing an anisotropy change in the YIG film	 • Frequency decrease by laser-induced heating the sample. No on-demand tuning of the magnon frequency. 	 • LLG-calculations only in the saturation regime

			 Experiments only in saturated regime 	
Ref [39] (previously [38])	Photoinduced strain wave in BiYIG	No magnons generated	No frequency shift	
Lucile PRL Ref. [38] (previously [37])	Laser-induced blueshift of magnon frequency	Polarization independent change of the anisotropy	 Experiments performed only in the saturated regime Only blueshift demonstrated, no on-demand frequency shift 	 Phenomenological theory in the saturated regime
Our work	On-demand optically induced magnon frequency-shift	 Polarization independent change of the anisotropy Effect does not depend on selective pumping of specific transitions of a dopant 	 Laser-induced tunability of the magnon frequency one order of magnitude more efficient (i.e. 4 mJcm⁻² needed for frequency tuning of 40%) 20 nm-thick sample, with magnetization one order of magnitude higher than in PRB 81 21440 (2010), saturated by a field of 125 mT Transition from the anisotropy-dominated to the field-dominated regime, i.e. on-demand optically induced frequency tuning. 	 Excellent Theory-experiment agreement in both field regimes (below and above saturation). Phenomenological theory, analytical theory and numerical simulations.

Actions Taken:

- We have added a sentence on page 2 citing the papers mentioned by the Reviewer and addressing their criticism:
“Alternatively the resonant pumping of specific electronic transitions from dopant ions²¹ and opto-magnetic effects²² have been explored. Moreover in several cases bulk materials and cryogenic conditions^{6,7,19} (to explore the phase diagram), were necessary. “
- We added sentences on page 3 commenting on the independence of the excitation mechanism from pump polarization:
“Other authors have reported a light-induced modification of both the magnitude and direction of the magnetic anisotropy²¹. This effect induces a dependence of the phase of the

coherent oscillations on the polarization of the pump beam. We rule out this scenario in view of Extended Data Fig. 2, which proves our mechanism to be pump polarization-independent consistently with the DECM picture.”

Comment 2

Another important aspect concerns the thermal nature of the employed mechanism. Heating can pose a significant obstacle to practical applications, as it limits the operational speed and leads to inefficient energy consumption. This issue is particularly critical in dielectric materials, where cooling is substantially slower than in metals. The observation that the frequency does not change significantly within the first nanosecond indicates that the cooling process is relatively slow. If the tuning of frequency is too slow it can be replaced by other approaches where an external magnetic field is varied to influence the spectrum.

Reply to comment 2

The Reviewer addresses also the important role of laser-induced heating in the optically-driven spin dynamics, in view of the dielectric nature of our sample. We agree with the Reviewer, that this point deserves a deeper discussion in our manuscript.

To address this aspect, first of all, we compare the laser-induced heating in our dielectric sample and with magnetic metals, which are alternative platforms for spintronics. The absorption coefficient of our BiYIG sample for the wavelength of the pump beam amounts to $7.5 \times 10^4 \text{ cm}^{-1}$ (estimated from measurements). The values of the absorption coefficient for the same wavelength in magnetic metals are: Ni - $1 \times 10^6 \text{ cm}^{-1}$, Fe - $1.3 \times 10^6 \text{ cm}^{-1}$, Co- $1.1 \times 10^6 \text{ cm}^{-1}$ (J. Phys. Chem. Ref. Data **38**, 1013 (2009)). Therefore, the energy dissipations in the case of BiYIG is more limited than in metals by at least one order of magnitude. An additional consequence of the free electrons in metals, promoting significantly more laser-heating, is a strongly reduced coherence time of spin waves, which would be detrimental for schemes of magnon-based wave-computing and for sources of spin waves with tunable frequencies.

The Reviewer properly observes that the cooling time is longer in dielectrics than in metals. We understand that this might appear as a limitation to the speed of the operation at first glance. However, it is actually beneficial when working with spin waves in the GHz frequency range: reconfiguration of the device on the nanosecond scale, which effectively means changing the functionality of the device itself, is desirable in order to perform the actual computation in a quasi-static setting while reconfiguring the device on longer timescales. For example, this nanosecond tunability enables the implementation of computing or transmission schemes with time multiplexing. Additionally, speed is not a concern when different bits are addressed locally in series as in the HAMR technology (Proceedings of the IEEE, **96**, 1810. (2008), J. Appl. Phys. **137**, 125111 (2025)). As our excitation scheme is local (we excite non-propagating magnons, i.e. spin waves at the centre of the Brillouin zone), different regions of the materials can be illuminated at the same time and the operation can be heavily parallelized following schemes suggested in the literature (Nat. Phys. **11**, 487–491 (2015), Rev. Sci. Instrum. **94**, 9 (2023)). Therefore, the cooling time of the material does not represent the main bottleneck for the operation of a device based on the concept introduced in this manuscript.

Actions Taken:

- Added sentence in the abstract addressing the questions of the Reviewer: “Our results show how an efficient manipulation of magnons can be achieved by light and provide perspectives for the realization of logic devices optically reconfigurable on the nanosecond timescale.”

Comment 3

The origin of the oscillations is explained as due to a reduction of the anisotropy field. However, in the geometry where the easy axis is strictly out of plane, the external magnetic field is in-plane, and the anisotropy field experiences only reduction (without change of direction), I would expect no signal. The authors mention such a scenario in the multidomain regime but do not resolve the obstacle. Furthermore, even in a monodomain sample, it is not clear which factors define the direction of the anisotropy field if the external field is strictly perpendicular to it.

Reply to comment 3

We thank the Reviewer for pointing out a lack of clarity in our description of the excitation mechanism. The key factor is the equilibrium position of the magnetization, which is determined by the interplay between the external magnetic field and the anisotropy of the material. If the applied field is not intense enough to saturate the magnetization, the magnetization lies at an angle respect to the plane of the garnet film (Fig. 2c, first panel). In this configuration a reduction of the anisotropy effectively results in a torque triggering the precession of the magnetization (Fig. 2c, second and third panels), because the equilibrium tilt angle is changed towards a new equilibrium between the perpendicular magnetic anisotropy (PMA) and the in-plane Zeeman energy.

If the field is intense enough to fully align the magnetization in plane, exactly 90 degrees away from the direction of the anisotropy ($\theta_{M,0} = \theta_H = 90^\circ$ in Fig. 2c), we fully share the expectation of the Reviewer: no coherent oscillations should be induced. However, the minor contribution from the magneto-crystalline anisotropy in BiYIG and the unavoidable misalignment of a few degrees between the external magnetic field and the direction $\theta_{M,0} = \theta_H = 90^\circ$, due to the manual setting of the magnetic field orientation, always results in a minor tilt of the magnetization, such that $\theta_{M,0}$ is never exactly 90° , even for a very large in-plane field. This allows us to photoinduce spin precession, although with significantly smaller amplitude than in the out-of-plane case (Fig. 2b).

We finally highlight that the direction of the anisotropy is mainly uniaxial, defined by the strain of the substrate, i.e., it is strictly perpendicular to the sample plane. This statement is supported by the SQUID characterization of our sample (Fig. R1, description in details in the reply to Comment 4) and is consistent with the literature of ultrathin rare-earth doped garnets (Nature Communications **9**, 3355 (2018), ACS Appl. Nano Mater **5**, 1023 (2022)):

Figure R1: SQUID magnetometry measurements. Magnetization VS external field SQUID magnetometry measurements for a) out-of-plane magnetic field and b), c) two different in-plane field directions. All data were acquired at different sample temperature. We note that “in-plane 1” and “in-plane 2” indicate two orthogonal directions in the plane of the sample parallel to the sample edges.

Our excitation mechanism cannot change the direction of the anisotropy, as such an effect would generate coherent oscillations whose phase would depend on the polarization of the pump

beam (PRB **81**, 214440 (2010)), see also our answer to Comment 5. Differently, the spin dynamics photoinduced in our case does not display any dependence on the polarization of the pump beam, as it can be seen in Fig. R2.

Figure R2: Pump polarization dependence of the spin dynamics. The probe beam with a photon energy of 2.9 eV was linearly polarized along the horizontal plane. The measurements were performed setting the excitation fluence to 3 mJcm^{-2} and applying an external magnetic field of 200 mT. This dataset rules out the possibility that the optical stimulus changes also the direction of the anisotropy in our case.

Actions Taken:

- We have added the figures reporting the SQUID measurements (Extended Data Fig. 4) and the pump polarization dependence (Extended Data Fig. 2) to the manuscript.
- We have added sentences on pages 2 and 3 to introduce these figures and to address the comment of the Reviewer.

“As a result, our specimen shows perpendicular magnetic anisotropy (PMA), as demonstrated by the SQUID measurements (Extended Data Fig. 1, see Methods).”

“We rule out this scenario in view of Extended Data Fig. 2, which proves our mechanism to be pump polarization-independent consistently with the DECM picture.”

Comment 4

Is the easy axis strictly perpendicular to the plane surface? Do the authors observe hysteresis in magnetization curves for in-plane magnetic field?

Reply to comment 4

The SQUID magnetometry (Fig. R1) has been measured with an MPMS-3 from Quantum Design. The raw data were corrected for parasitic signals of the substrate and sample holder to extract the signal stemming from the BiYIG thin film. The out-of-plane magnetisation VS magnetic field hysteresis loop (Fig. R1 left) reveals a perpendicular magnetic anisotropy (PMA), which persists even at elevated temperatures with a field of less than 30 mT needed to fully saturate the film. The saturation magnetization decreases with increasing temperature.

We do not observe hysteresis that can be attributed to the BiYIG thin film by applying the external magnetic field along the two in-plane directions (Fig. R1 b) and c)). This indicates that the magnetization is not stabilized in the sample plane, i.e., that the system does not possess an in-plane easy axis. This observation is consistent with the presence of a strong perpendicular magnetic anisotropy, where the easy axis is oriented out-of-plane, in accordance with the

literature of BiYIG (Nat. Commun. **9**, 3355 (2018), Phys. Rev. Lett. **127**, 077203 (2021)). This aspect is also apparent in the MOKE data in Fig. 1c of the main manuscript.

Actions Taken:

- We have added the SQUID measurements (Extended Data Fig. 4) to demonstrate that the sample exhibits exclusively perpendicular magnetic anisotropy (PMA) with negligible in-plane anisotropy.
- We have added a sentence on page 2 to introduce this figure.

“As a result, our specimen shows perpendicular magnetic anisotropy (PMA), as demonstrated by the SQUID measurements (Extended Data Fig. 1, see Methods).”

Comment 5

Is it only reduction of anisotropy or there are changes in its orientation? The authors demonstrate that their signals do not depend on excitation polarization (pump). It is however not clear what happens when the direction of magnetic field changes in-plane with respect to crystallographic axes.

Reply to comment 5

The question of the Reviewer addresses an important aspect of our work, as it distinguishes our observation from the literature. We agree that the mechanism needs to be explained more comprehensively than what was provided in the original version of the manuscript. The literature (PRB **81**, 214440 (2010)) demonstrates that an optical modification of the orientation of the anisotropy results in coherent oscillations, whose phase depends on the polarization of the pump beam. In a nutshell, it is a polarization-dependent effect. Our situation is different, as the data shown in Fig. R1 reveal the spin dynamics to be independent of the polarization of the pump beam. This observation is therefore compatible with the idea that pump pulses suppress the anisotropy without changing its orientation.

With our present set-up, we cannot modify the direction of the magnetic field in the plane of the sample. However, changing the magnetic field direction in the plane of the sample is expected to provide only minor changes to the observed signal. In fact, the in-plane anisotropies in BiYIG (a fraction of the magneto-crystalline anisotropy of $\approx 600 \text{ J m}^{-3}$ (J. Appl. Phys. **31**, S376 (1960)) are usually neglected, as they are much weaker than the perpendicular anisotropy ($\approx 11 \text{ kJ m}^{-3}$ in our sample) of magnetostatic origin. This is confirmed by the SQUID measurements shown in Fig. R1, right panel.

Actions Taken:

- Added sentences on page 3 addressing this point and the measurements as a function of the pump polarization to the manuscript (Extended Data Fig. 2):
“This effect induces a dependence of the phase of the coherent oscillations on the polarization of the pump beam. We rule out this scenario in view of Extended Data Fig. 2, which proves our mechanism to be pump polarization-independent consistently with the DECM picture.”

Comment 6

What is the role of ultrathin material in this study? What is the size of domains?

Reply to comment 6

Our sample is grown via a high-temperature off-axis radio-frequency (rf) magnetron sputtering process, and is subject to tensile strain on its GSGG substrate. The lattice parameters of our $\text{Bi}_{0.8}\text{Y}_{2.2}\text{Fe}_5\text{O}_{12}$ and GSGG substrate are 1.245 nm and 1.255 nm, respectively, which corresponds

to a mismatch of -0.8 %. Using this method, the tensile strain in BiYIG can induce perpendicular magnetic anisotropy (PMA), provided that the thickness of the sample is smaller than ≈ 100 nm to prevent strain relaxation (see also Nat. Commun. **9**, 3355 (2018), ACS Appl. Nano Mater **5**, 1023 (2022)). The PMA enabled by such thin strained samples is key for our scheme of laser-induced tuning of the frequency of coherent magnons, as it determines the ability of laser pulses to either weakly perturb or completely suppress the PMA, and enables to arbitrarily red- or blue-shift the magnon frequency.

The size of the domains in our films lies in the 1-5 μm range, as shown in the MOKE imaging data shown in Fig. 1a. The average size of the magnetic domains in perpendicular magnetic anisotropy films is indeed strongly influenced by the film thickness (Philips Red. Repts. **15**, 7 (1960), J. Magn. Magn. Mater. **128**, 111 (1993)).

Suggested actions Taken:

- We have added a sentence on this aspect in the manuscript.

Comment 7

What is the origin of magnon damping? How strong is the effect of heating on damping?

Reply to comment 7

The magnon damping is due to interactions between spins and the environment, primarily the lattice. Following the advice of the Reviewer, we explore the effect of the laser-heating on the damping, by measuring the lifetime of the magnons as a function of the laser fluence. Here below we show the results for the lifetime τ and, equivalently, for the effective damping ($\alpha_{eff} = 1/(2\pi\tau f)$). We note that the lifetime is not equivalent to but shorter than the Gilbert damping, as several line broadening mechanisms (especially inhomogeneous due to scattering channels) are photoinduced by the pump beam.

Figure R3: Lifetime and effective damping. Lifetime time (left) and effective damping (right) as a function of the pump fluence. Measurements were performed under an external magnetic field of 200 mT.

The effective damping increases linearly as a function of the fluence up to the value of 5 mJ/cm^2 . Above this value a saturation regime can be recognized in the data, taking into account the error bars. We interpret this result in terms of a laser-induced population of incoherent phonons, which by scattering with the spin waves, shorten their lifetime. However, this interaction channel seems to encounter a bottleneck in correspondence of the fluence value of 5 mJ/cm^2 . Since the damping already takes into account the modification of the frequency (see definition above), the results shown in Fig. R3, display a change of the quality factor of the magnetic resonance. A detailed and microscopic understanding of the interaction leading to the laser-induced modification of the effective damping of coherent spin waves is beyond the scope of the current manuscript. However, we performed calculations of the effect of the ellipticity of the spin precession on the damping time, which turns out to be negligible for a uniaxial anisotropy field of 220 mT. This effect

becomes even smaller, if the anisotropy field is reduced, which is exactly what happens once the sample is illuminated by the pump beam.

Suggested Actions Taken:

- Added Fig. R3 above to the manuscript and a sentence describing these conclusions
“We observe that the magnon damping depends on the laser fluence (see Methods and Extended Data Fig. 6 b)). It increases linearly up to 5 mJ/cm^2 before saturating. This behavior is attributed to laser-induced incoherent phonons that scatter with spin waves, reducing their lifetime until a bottleneck occurs near this fluence. A microscopic explanation is beyond the scope of the present manuscript.”

Reply to Reviewer #2:

This manuscript reports experimental studies of the photo-excitation of coherent magnons by visible light pulses in films of bismuth-substituted yttrium iron garnet (Bi:YIG) under an external magnetic field applied in the film plane. The magnons are excited by a pump beam with varying fluence of photons with energy 2.4 eV by means of a mechanism known as displacive excitation of coherent magnons (DECM), in the equilibrium orientation of the magnetization is displaced, initiating a precessional motion characterizing a zone-center magnon. The magnon detection is made by a probe beam of photons with energy 2.9 eV tuned to a resonance of the magneto-optical spectrum of Bi:YIG. The detected probe beam exhibits a transient oscillating decaying response from which the magnon frequency is extracted. The authors claim that their results show how efficient manipulation of magnons can be achieved by light and provide perspectives for the realization of tunable spin-wave generators and reconfigurable logic devices. The results are very interesting, the experiments and theoretical interpretation are presented in detail, and the paper is well written.

Comment 1

In my opinion the paper will attract attention of the magnetism and spintronics community and deserves to be published in Nature Communications. The only minor suggestion I have is that in the title and text the word tune is better than shift. I understand that shift would apply to a magnon excited with a certain frequency and while it is propagating an external action changes, or shifts, its frequency, as it was shown a long time ago in Frequency Conversion of Spin Waves in Pulsed Magnetic Fields, Applied Physics Letters 10, 184 (1967).

Reply to comment 1

We thank the Reviewer for the positive assessment of our work. We agree with the suggestion and modify the title accordingly.

Actions Taken:

- We have changed the title of the manuscript and replaced the word “shift” with “tuning” throughout the text

Reply to Reviewer #3:

The authors show the red- and blue-tunability of the magnon frequency in a strained 20 nm BiYIG thin single-crystal. They control the optical excitation fluence and the magnitude of the externally applied magnetic field (to modify the magnetic state before photoexcitation) to transiently alter the uniaxial magnetic anisotropy and magnetization direction, which initiates the precessional motion at FMR. The authors find that in the low-fluence regime, the response is dominated by anisotropy, while in the high-fluence regime, it is dominated by the external field. Atomistic spin dynamics simulations qualitatively capture the experimental findings.

Measurements, data analysis, and simulations follow standard protocols and equations; hence, I find them satisfactory.

The interesting angle the authors use here is the external field regime where magnetization is not saturated and a multi-domain state is present, which leads to red- and blue-shifted responses in the same material, which is not shown before and has the potential for the realization of tunable spin-wave generators and reconfigurable logic devices.

Comment 1

As the authors rightly point out, in a multi-domain state, a coherent signal is not expected due to the random orientation of domains, but a coherent signal is observed here. I am not quite sure why a coherent state is observed. Is the multi-domain state due to an external field that is not quite random, or is the same type of domains dominating the response? The authors must address this point conclusively by mapping the orientation of multi-domain state and also address why it leads to a coherent signal.

Reply to comment 1

The Reviewer raises a very good point and we fully understand their expectation, as detecting a coherent signal in the multidomain state was originally surprising for us as well. We have understood this result considering that an in-plane field biases the local oscillation axis of the magnetization to a specific direction. Consequently, coherent signal from different domains with the same magnetization direction dominates on the signal generated by domains, in which the magnetization is differently oriented. To tackle this point experimentally, we have performed new measurements as a function of the magnetic field, especially addressing the multi-domain regime, down to a vanishing field. The evaluation of the amplitude (fit-parameter a in M10 in the main manuscript) of the magnon oscillations, measured as a function of the in-plane field (Fig. R4a) confirms our statement: the field biases the local axis of the magnetization to a specific direction. It can be observed that as the in-plane field is reduced, the signal amplitude decreases and approaches zero for 0 mT (Fig. R4b)), as anticipated by the Reviewer. This is also shown by atomistic simulations performed for decreasing magnetic fields down to 1mT. In agreement with the experimental results, simulations show how the signal amplitude decreases approaching zero as the in-plane field is reduced (Fig. R4c)).

Figure R 4: External field dependence of oscillation amplitude and frequency. a) Selection of time-resolved transient probe polarization rotations for different external fields ranging from 180 to 0 mT. The data are horizontally offset for better visibility. Oscillations are absent only at 0 mT. b) Amplitude of the magnetization oscillation for different external magnetic fields obtained experimentally and c) from atomistic simulations. As the field approaches 0 mT the amplitude vanishes. d) Corresponding oscillation frequencies to the presented time-resolved measurements and e) the frequencies obtained from simulations. Pump and probe beams were horizontally polarized. The excitation fluence in these measurements was set to 0.5 mJcm^{-2} in experiments and 2 mJcm^{-2} in simulations.

Our results are also in agreement with the literature that demonstrates that two main modes should be measurable in the multidomain state (Nature Nanotechnology **14**, 691–697 (2019), J. Phys.: Condens. Matter **29**, 465803 (2017) or Phys. Rev. B **89**, 024411 (2014)), originating either from the domain volume or from the dynamics of the walls. The frequency of the domain volume mode decreases as the applied magnetic field increases. This trend is consistent with our observation, suggesting that our scheme is sensitive only to the domain volume mode. We believe that the dynamics of the wall cannot be monitored, because the volume of the walls is much more limited than the volume of the domains. Our experimental method is sensitive to an average response throughout the sample volume, so the response of the domain dominates.

Unfortunately, in our time-resolved scheme we cannot address different isolated domains within the multidomain states. We cannot perform time-resolved magneto-optical imaging. We could image the domains statically, relying on a commercial magneto-optical microscope.

Actions Taken:

- We have added a sentence on page 4, a paragraph in the Methods section and Fig. R4 (Extended Data Fig. 8).

“We understand the data in multi-domain state considering that the in-plane field biases the local oscillation axis of the magnetization to a specific direction. Consequently, coherent signal from different domains with the same magnetization direction dominates on the signal generated by domains, in which the magnetization is differently oriented. The magnetic field-dependence of the magnon amplitude (Extended Data Fig. 3(b)) confirms this statement, showing that as the in-plane field is reduced, the signal amplitude decreases and approaches zero for a vanishing field. Additional measurements performed sweeping the field from -200 mT to 200 mT and back again to -200 mT (Extended Data Fig. 4) reveals that the sign of the

field and the direction of the field sweep have no systematic detectable effect on the frequency and amplitude of coherent magnons.”

Comment 2

The authors have also not addressed what happens when we follow the hysteresis curve from -200 mT to 200 mT and then cycle back from 200 mT to -200 mT. Is the response different?

Reply to comment 2

To address this question, we performed additional measurements. Figure R5 shows the oscillation frequency (panels a) and d)) and amplitude (panels b) and e)) of the photoinduced coherent magnons measured by following the hysteresis loop, as requested by the Reviewer. The loop was first traced from -200 to 200 mT (blue) and then back (red) for fluences of 0.5 and 8 mJcm⁻². The data do not display systematic changes, depending on the direction of the field sweep and the sign of the applied magnetic field. This result is consistent with the numerical simulations performed: sweeping the magnetic field from negative to positive values does not induce a noticeable systematic difference (c)).

Figure R 5: Frequency and oscillation amplitude for full hysteresis loop. The left upper panel indicates the sweep direction. The corresponding data in a), b), d) and e) are color-coded accordingly: data from -200 to 200 mT are shown in blue, and data from the return path are shown in red. a), d) Oscillation frequencies of the photoinduced coherent magnons and b), e) corresponding normalized amplitudes of the oscillation for the full hysteresis loop for the fluences 0.5 and 8 mJcm⁻². Pump and probe beams were horizontally polarized. c) Simulation results for external field sweep from positive (green) to negative (purple) fields for an excitation fluence of 2 mJcm⁻².

Actions Taken:

- We have added a sentence on page 4, a paragraph in the Methods section and Fig. R5 (Extended Data Fig. 9).
“Additional measurements performed sweeping the field from -200 mT to 200 mT and back again to -200 mT (Extended Data Fig. 4) reveals that the sign of the field and the direction of the field sweep have no systematic detectable effect on the frequency and amplitude of coherent magnons.”

Comment 3

The authors must carry out detailed sample characterization, such as magnetization vs T data, strain value, and magnetic moment on both sublattices. Currently, values are used from the literature, but since these values can potentially impact reproducibility, the authors must show these measurements.

Reply to comment 3

We performed SQUID measurements demonstrating that our samples indeed show PMA (Fig. R1). From these data, we were able to extract the saturation magnetization for values of temperature ranging from 300 K to 380 K. This allows for a description of the laser-induced demagnetization on the basis of our SQUID measurements (Fig. R5), instead of relying on parameters from the literature, as we did in the previous version of the manuscript.

Figure R6: Estimation of laser-induced demagnetization. Temperature dependent magnetization curve with modified Bloch function as a guide to the eye of the magnetization change due to laser-induced heating. Data extracted from the SQUID hysteresis loops. Inset: Calculated laser-induced demagnetization from the temperature dependence of the magnetization for different excitation fluences.

The subsequent data fitting and extraction of the anisotropy can thus also be performed using the measured SQUID magnetometry data. We plot a function describing the temperature dependence of the magnetization incorporating only the first leading term (Phys. Rev. B **95**, 214423 (2017)) as a guide to the eye, which is a modified function of the conventional Bloch model. This requires slight modifications of formulas M(11) and M(13) in the main text, as the temperature dependence of the magnetization is referenced to the measured magnetization at 300 K. The modified Bloch function is then given by:

$$\frac{M(T)}{M(T = 300\text{K})} = M_0 \cdot (1 - aT^{5/2}), \quad \text{M(11)}$$

with the parameters $M_0 = 1.205$ and $a = 1.039 \times 10^7 \text{ K}^{-5/2}$. The demagnetization induced by laser heating can be calculated using

$$\Delta M(F) = M_0(1 - a(T + \Delta T)^{5/2}) - M_0(1 - aT^{5/2}). \quad \text{M(13)}$$

With the modified equations, the subsequent fitting of the FMR curves for the different excitation fluences can be performed, yielding the following anisotropy modification:

Figure R 7: FMR data fitting and determination of the anisotropy. Frequencies from the time domain and the corresponding fits with Eq. M(2) to determine the anisotropy for different fluences for external fields in saturation using the temperature/fluence dependence of the magnetization. The values of the anisotropy determined from the fit to the data with Eq. M(2) is shown in the lower panel.

Actions Taken:

- Add the figures R1 and R4 to present our own measured data and to support reproducibility.
- Add updated fits R5 using the new fit parameters of demagnetization.

List of changes

1. Title: "Tuning" instead of "shift"
2. Replaced the word "shift" with "tuning" throughout the text
3. Last sentence in the abstract: "Our results....on the nanosecond timescale"
4. Page 2: "Alternatively the resonant pumping of specific electronic transitions from dopant ions²¹ and opto-magnetic effects²² have been explored."
5. Page 2: "...), as demonstrated by the SQUID measurements (Extended Data Fig. 4, see Methods)."
6. Extended Data Fig. 4 added
7. Page 3: "Other authors ...consistently with the DECM picture".
8. Page 4: "We understand the...coherent magnons".
9. Page 4: "We observe...present manuscript."
10. Extended Data Fig. 8 and 9 added.

Reply to Reviewer #1:

Comment 1

As a minor point, it remains somewhat unclear to me why the authors were able to achieve such low thresholds and outstanding tuning characteristics in their work. Is this mainly related to particular properties of the samples they used? It would be beneficial to include a brief explanation in the text to clarify this point.

Reply to comment 1

We thank the reviewer for pointing out the imprecision in our description. The observed behaviour can be attributed to the inherently low magnetic damping of Bi:YIG, its enhanced magneto-optical response, and the low thermal conductivity of the GSGG substrate, which confines the laser-induced heating. These factors collectively distinguish the system from conventional YIG/GGG heterostructures. For reference, a comparative overview of substrate thermal conductivities is provided in Table 5 of Reference 24, which we have added to the manuscript. The growth method employed to fabricate our samples is extensively described in the added reference [23].

Actions Taken:

- Added sentence addressing the questions of the Reviewer:
“The observed behavior arises from Bi:YIG’s low damping, strong magneto-optical response, and the low thermal conductivity of the GSGG substrate, distinguishing it from conventional YIG/GGG systems^{23,24}. ”

List of changes

1. Added the sentence addressing the Reviewers question on page 2